Optimizing the trainable B-COSFIRE filter for retinal blood vessel segmentation

Badawi Sufian A. Sufian.badawi@seecs.edu.pk
Fraz Muhammad Moazam
School of Electrical Engineering and Computer Science, National University of Sciences and Technology , Islamabad , Pakistan
Huisman Henkjan
Electronic publication date: 2018 Nov 13
Publication date: 2018
Volume: 6
Electronic Location ID: e5855
Received 2018 Mar 20; Accepted 2018 Sep 28
Copyright: ©2018 Badawi and Fraz
Copyright year: 2018
Copyright holder: Badawi and Fraz
License: This is an open access article distributed under the terms of the Creative Commons Attribution License, which permits unrestricted use, distribution, reproduction and adaptation in any medium and for any purpose provided that it is properly attributed. For attribution, the original author(s), title, publication source (PeerJ) and either DOI or URL of the article must be cited.
License URL: https://creativecommons.org/licenses/by/4.0/

Keywords: Retinal blood vessels, B-COSFIRE, Retinal images, Computer Aided Diagnosis (CAD), BCOSFIRE

Funding: The authors received no funding for this work.

==============================
Segmentation of the retinal blood vessels using filtering techniques is a widely used step in the development of an automated system for diagnostic retinal image analysis. This paper optimized the blood vessel segmentation, by extending the trainable B-COSFIRE filter via identification of more optimal parameters. The filter parameters are introduced using an optimization procedure to three public datasets (STARE, DRIVE, and CHASE-DB1). The suggested approach considers analyzing thresholding parameters selection followed by application of background artifacts removal techniques. The approach results are better than the other state of the art methods used for vessel segmentation. ANOVA analysis technique is also used to identify the most significant parameters that are impacting the performance results (p-value ¡ 0.05). The proposed enhancement has improved the vessel segmentation accuracy in DRIVE, STARE and CHASE-DB1 to 95.47, 95.30 and 95.30, respectively.

Introduction

The analysis of shape, appearance, tortuosity and other morphological characteristics of blood vessels in human retinal images are critical diagnostic indicators to eye diseases. Various ophthalmic and systemic diseases including diabetic retinopathy, age-related macular degeneration, hypertensive retinopathy, arteriolar narrowing, and arteriosclerosis (Kanski et al., 2011). The association of abnormalities in retinal vasculature with cardiovascular diseases has been reported in the literature  (Wong et al., 2002). Therefore, accurate segmentation of retinal blood vessels can be considered as the first step in the development of an automated system for diagnostic retinal image analysis.

The segmentation of blood vessels in the retina is a challenging problem. In the retinal image captured from fundus camera, the blood vessels emerge from retinal Optic Disc with their branches spread across the retina. The retinal vasculature is comprised of arterioles and venules, appears to be piecewise line-shaped and its width is variable across the vessel length (Abràmoff, Garvin & Sonka, 2010). The cross-sectional gray level intensity profile of retinal vessel approximates the Gaussian-shaped curve or a mixture of Gaussian in case central vessel reflex is present. The central vessel reflex appears as a strong reflection across the vessel centerline and is more evident in the arterioles of retinal images of children due to the difference in oximetry level than that of adults (Fraz, Basit & Barman, 2013). The crossover of different arterioles and venules branches and its branch-points further complicates the vessel profile model. There may be a number of pathological structures present in the retina which includes hemorrhages, exudates, drusens, cotton-wool spots and microaneurysms, etc. The intensity profiles of some of these pathologies may resemble that of the boll vessels. Besides variation in contrast, uneven illumination during image capture, low quality of the captured retinal image and the presence of pathologies are the further added challenges for the development of robust automated retinal vessel segmentation methodologies (Fraz & Barman, 2014).

Several approaches have been proposed in the literature for the segmentation of retinal vasculature. A comprehensive review of retinal vessel segmentation methods is also available in (Fraz et al., 2012b). These methods can be classified into two major categories, supervised and unsupervised methods.

The supervised methods compute pixel-wise features and use the labeled ground truth data to train the classifiers. The prerequisite is the availability of labeled data which is difficult to obtain in case of medical images. In this regard, various classifiers have been used including k-Nearest Neighbor classifier (Staal et al., 2004), Gaussian Mixture Model (Soares et al., 2006), Support Vector Machine (Ricci & Perfetti, 2007), and ensemble classifier of Decision Trees (Fraz et al., 2012c), Bayesian classifier in combination with multi scale analysis of Gabor wavelets (Soares et al., 2006), Neural Network that can be classified as shallow (Marín et al., 2011) or deep learning that creates a real advancement in computer vision by introducing the improvement such as the rectified linear units with the convolutional neural networks (CNNs) (Long, Shelhamer & Darrell, 2015; Chen et al., 2014; Krizhevsky, Sutskever & Hinton, 2012), a variety of CNNs-based methods have been introduced for the vessel segmentation of the retinal images. Li et al. (2016) proposed a deep neural network to model it as a cross modality transformation problem. On the other hand, Fu et al. (2016) achieved the vessel segmentation as a CNN combined with a fully-connected Conditional Random Fields (CRFs). Maninis et al. (2016) have introduced vessel and optic disk segmentation using CNN. Dasgupta (Dasgupta & Singh, 2017) implemented a pixel wise binary classification of the retinal image using batches of 28 ×  28 pixels each. Orlando, Prokofyeva & Blaschko (2017) proposed for the vessel segmentation a trained discriminative conditional random field model.

In contrast the unsupervised method does not use classifiers, but relies on the application of matched filtering that is based on techniques of kernel matching and special filtering that relies on linear operations using predefined templates (Azzopardi et al., 2015; Cinsdikici & Aydın, 2009; Fraz et al., 2012a; Hari, Raj & Gopikakumari, 2017; Sofka & Stewart, 2006), vessel tracking (Yin, Adel & Bourennane, 2012), model fitting (Al-Diri, Hunter & Steel, 2009), and mathematical morphology (Fraz et al., 2012b; Fraz et al., 2012a; Wisaeng & Sa-ngiamvibool, 2018), from the non-supervised methods Combination Of Shifted Filter Responses (Azzopardi & Petkov, 2013b), Rotating Derivative of Left invariant (Zhang et al., 2016), Fussy convergence (Hoover, Kouznetsova & Goldbaum, 2000), local Adaptive thresholding and mathematical morphology citepmendonca2006segmentation, multi-scale analysis (Martinez-Perez et al., 1999), rotation-invariant line operator and linear kernel support vector machine (SVM) (Lau et al., 2013), Ribbon of Twins (Al-Diri, Hunter & Steel, 2009), and multi-concavity modeling approach (Lam, Gao & Liew, 2010).

Recently, a trainable template matching method for vessel segmentation was introduced (Strisciuglio & Petkov, 2017; Strisciuglio, Azzopardi & Petkov, 2017; Azzopardi & Petkov, 2013b; Azzopardi et al., 2015; Cinsdikici & Aydın, 2009; Fraz et al., 2012a; Hari, Raj & Gopikakumari, 2017; Sofka & Stewart, 2006). It segments the tree musculature via parameters configuration. They manage to detect the shape of vessels by prototyping two patterns. These patterns are in the form of bar and half bar. They configure the geometric features of the pattern to detect the vessels in the retina and segment it. However it needs a manual configuration for the pattern detection which is prone to errors, moreover, it is noticed that there are some background artifacts that may increase the false positive ratio which in turn will affect the accuracy and precision ratio. Thus to improve the results of the B-COSFIRE method we hypothesize that we need to optimize the thresholding parameter values and background artifacts removal mechanism.

This work aims at enhancing the parameter configuration of trainable filter that achieves a higher vessel segmentation performance on publicly available datasets DRIVE, STARE, and CHASE-DB1.

The rest of this paper is organized as follows: ‘Trainable COSFIRE Filter’ an overview of the trainable COSFIRE filter. ‘Trainable B-COSFIRE Filter’ is the overview of B-COSFIRE filter. ‘Improving B-COSFIRE Method’ is describing the improved B-COSIRE filter. ‘Experimental Evaluation’ is explaining experimental evaluation. ‘Results’ discusses results. ‘Discussion’ presents the discussion. While the final section discusses conclusion and future work.

Trainable COSFIRE filter

Combination Of Shifted Filter Responses (COSFIRE) is an unsupervised pattern detection method in computer vision. It is based on the computational model ‘Combination of Receptive Fields’ (CORF) in the visual cortex of the brain (Strisciuglio, Azzopardi & Petkov, 2017; Azzopardi & Petkov, 2012). COSFIRE filter is invariant to scale, rotation, shift and reflection transformations (Azzopardi & Petkov, 2013b). The learning and recognition are comparatively fast with high accuracy and shape selectivity (Cadieu et al., 2007). It is used for the detection of contour-based patterns (as shown in Fig. 1A).

Figure 1 (A) DoG functions showing the contours of each function; (B) illustration of the vessel as a bar and identifying the five points to design the B-COSFIRE prototype.

The applications of COSFIRE in computer vision include keyword detection in handwritten text (Azzopardi & Petkov, 2014), complex scene objects identification (Azzopardi & Petkov, 2014), color based object recognition (Gecer, Azzopardi & Petkov, 2017), gender recognition using facial features (Azzopardi, Greco & Vento, 2016), edge detection (Azzopardi & Petkov, 2012), handwritten digits recognition (Azzopardi et al., 2016), traffic signals recognition (Azzopardi et al., 2016), and detection of vessels bifurcations in retinal images (Azzopardi et al., 2016).

Difference of Gaussian (DOG) filter is used for generating the combination of filter responses. The COSFIRE method can be classified as a combination of shifted filter responses, key point detection, and pattern recognition. It is a trainable filter and combines the responses from a group of DoG filters (represented by blobs).

Trainable B-COSFIRE filter

Bar-Combination of Shifted Filter Responses (B-COSFIRE) (Azzopardi & Petkov, 2013b) is based on COSFIRE filter. It is an extension of a combination of the Shifted Filter Responses. It is also unsupervised vessel segmentation method as COSFIRE method.

B-COSFIRE is trainable to be selective for the symmetric and asymmetric bar-shaped patterns. It takes its input as five blobs from the DoG filter on identified points at a specific distance from the center of the kernel (as shown in Fig. 1B). Also, being a template matching filter, it achieves the filter response by extracting the dominant orientation that has the concerned features and its geometric arrangements, Its output is computed as the weighted geometric mean of blurred and shifted responses of the DoG.

The applications of B-COSFIRE in computer vision are: detection of roads, cracks, rivers, tile edges, land plots and plant leaf’s nerves (Strisciuglio & Petkov, 2017), and detection of vessels in retinal images (Strisciuglio, Azzopardi & Petkov, 2017).

The point locations are identified and enumerated in the bar and half bar via an automatic configuration process as shown in Figs. 2 and 3. The filter is described as ‘trainable’ and is configured automatically by detecting a given pattern of interest for the retinal vessels. There are two patterns; the first one is for the detection of internal part of the vessel while the second one is used to detect the end of the vessel. These two kernels are rotated in twelve orientations in order to cover all probable vessel directions. As a result, it forms a filter bank of 15° rotation of the filter that is good enough for optimal detection.

Figure 2 Illustration of B-COSFIRE prototype and the generated B-COSFIRE filter; (A) DoG filter; (B) automatically created a B-COSFIRE filter for the designed prototype; (C) bar-shaped prototype.

Figure 3 B-COSFIRE Bar and half bar filter prototype patterns.

(A) Bar-shaped prototype pattern. (B) Half bar-shaped prototype pattern. (C) Illustration of the designed DoG filter combinations to generate the pattern blobs for the B-COSFIRE filter.

B-COSFIRE filter uses the DoG filter for the detection of patterns (as shown in Fig. 3). However, Gabor filter used by Strisciuglio, Azzopardi & Petkov (2017) and Azzopardi & Petkov (2013a) and first-order derivative used by Fraz, Basit & Barman (2013) are also good alternatives that can be considered for the same task. B-COSFIRE filter is a trainable filter which combines the responses of DoG by multiplication of the response outputs. The configuration is comprised of orientations that are illustrated by ellipses, and blurring function is illustrated by 2D Gaussian blobs as shown in Fig. 4.

Figure 4 (A) Difference of Gaussian (DoG) 3D function and the blob generated from its contours. (B) The (DoG) blob. (C) Multi-scale of DoG 3D function and the created blobs.

B-COSFIRE filter is composed of the following steps:

1. For DoG filter responses (as shown in Eq. (1)) convolve DoG filter on the input image as in Eq. (4).

2. Blur the gained responses of the above DoG filters as in Eq. (2).

3. Shift the generated blurred responses to the direction of the filter center as in Eq. (3).

4. Getting the B-COSFIRE filter response by computing weighted geometric mean rs(x,y) as in Eq. (6).

The details of the above-mentioned steps can be found in Azzopardi & Petkov (2013b). (1) DoGσ=def12πσ2 expx2+y22σ2−12πσ2 expx2+y220.5σ2.

The given location (x, y) represents the center of DoG function as in Eq. (1) where sigma is the standard deviation (SD), and it represents the intensity profile spread. The SD of the inner Gaussian function is 0.5. (2) σ′=σo+αρi.

The next step is to blur the gained responses by applying Eq. (2), and the third step is to shift the blurred DoG responses with the help of shift vector Eq. (3), achieved with shift vector.

(3) △xi,△yi=−ρi cosθi−ρi sinθi

(4) Cσ=def|I⋆DoGσ|+.

For a given intensity distribution I(x′, y′) of an image I, the response C(σ)(x, y) of a DoG filter with a kernel function DoGσx−△x−x′,y−△y−y′. If the result of the convolution is negative it is replaced with 0 as shown in Eq. (4).

(5) Sσi,ρi,ϕix,y= maxx−△x−x′,y−△y−y′DoGσ′x′,y′

(6) rsx,y=def|∏1sSσi,ρi,ϕix,yωi1 ∑1sωi|t

where (7) 0≤t≤1,ωi=expρi22τ2,τ=13maxρi1|s|.

The filter threshold (t) is used to suppress the output of B-COSFIRE filter Eq. (6). The resulting B-COSFIRE filter is achieved by applying a number of responses that are orientation-selective and arranged around the point (x, y).

Improving B-COSFIRE method

To improve the results of the B-COSFIRE method we hypothesized that we need to optimize the thresholding parameter values and background artifacts removal mechanism. As it was noticed that there is some background artifacts that may increase the false positive ratio which in turn will affect the accuracy and precision ratio. In this work, we have selected three parameters for optimization (preprocessing threshold, filter threshold, and background artifact size), the ‘pre-processing threshold’ parameter is used to suppress the input filter responses that are less than a specific value defined, while the second parameter ‘filter threshold’ shown in Eq. (6) is used to suppress the response of B-COSFIRE filter. If the response value is less than the max response of ‘filter threshold’ and the ‘background artifact size’ is less than the size of the connected component, then it will be deleted. Moreover, three background artifacts removal algorithms are applied here in this experiment to get an enhanced output. The first background artifacts removal mechanism used is called Background artifacts Filtering algorithm and it focuses on removing all small size noisy objects that are disconnected and leaves only the vascular tree. The second is known as K-Median it uses the K-median clustering algorithm for removing the background artifacts and the third algorithm is known as the black and white artifact clearance that is used to remove small objects and fill small holes.

To evaluate the proposed approach, each dataset is divided into two equal sets, one for evaluation and the other for testing. The evaluation set is used as input to identify the best parameters. The training images of DRIVE dataset are used for evaluation, while for STARE and CHASE-DB1, we split each of them into two subsets, one for evaluation and, the other for testing of the proposed method.

It is pertinent to highlight the simplicity of the employed optimization technique, which is inspired from the random hyper parameter optimization search in Bergstra & Bengio (2012). We have defined a limited search range of parameters which are shown in Table 1. Afterwards, we have conducted multiple experiments to identify the best combination of parameter values. For each experiment, the sensitivity, specificity, and accuracy are calculated and documented. The parameter optimization for each dataset is illustrated in Fig. 5. Initially, we performed a number of experiments on the evaluation sets with the aim of determining the best combination of parameters for B-COSFIRE. Afterwards, Grid search method is employed for the purpose of identifying the combination of parameters that gives the optimal score of accuracy, sensitivity and specificity on the training images.

Table 1 Range of parameter values for the optimization of results.

Parameter	Range start	Range end	
Preprocessing thresould	0.1	0.6	
Filter threshold	25	50	
Background artifact size	0	48	

Figure 5 Schematic representation of our proposed parameter optimization.

We let the values of the parameters be incremented to cover all the studied parameters search space till the optimized values were achieved.

From another angle, to illustrate the benefits of our proposed approach. It is suggested to use one way analysis of variance ANOVA to compare statistical significance of the different values of each parameter on the results of the vessel segmentation.

Experimental Evaluation

This section briefly explains the details about the datasets used for conducting these experiments and showing the performance measures used on each of these datasets.

Materials

The optimization experiments are performed on three publicly available datasets (DRIVE, STARE, and CHASE-DB1) to find the optimal parameters.

DRIVE dataset

DRIVE dataset (Niemeijer et al., 2004) is a standardized set of fundus images used for benchmarking the effectiveness of vessel segmentation and classification. These images were captured in Netherlands for the purpose of screening the diabetic retinopathy. A total of 400 diabetic subjects data is collected between the age group of 25–90 years. All images are compressed with JPEG. Canon CR5 3CCD non-mydriatic camera was used for capturing these images. The resolution of each image is 768 by 584 pixels having 8 bits per pixel. A set of 20 images in each training and test datasets. The dataset includes manually prepared ground truth images and masks that are made by two experts.

STARE dataset

STARE (Hoover, Kouznetsova & Goldbaum, 2000) dataset was used for blood vessel segmentation. It is composed of 20 retinal images 10 of them have pathologies. The image capturing is achieved using a TopCon TRV-50 fundus camera. Each image has a resolution of 605 by 700 pixels with 8-bit RGB channels. Two sets of manually segmented images are prepared by two different observers.

CHASE-DB1 dataset

CHASE-DB1 (Fraz et al., 2012c) dataset is collected in England for Child Heart and Health Study. It contains 28 pairs of high-resolution fundus images, the resolution of images is 1,280 by 960 pixels. A total of 14 children left and right eyes images were captured with a 30° field of view. The Nidek NM-200-D fundus camera is used for obtaining these images. It is more prone to bias because all the images are paired with the same person.

Performance measures

For performance measures selection used in the experiments, let’s assume that the input image is “I” and assume the B-COSFIRE segmented image be “Y”. To decide whether the segmented output “Y” is correct or not, it is compared with the corresponding manual hand labeled image, customarily called Ground Truth label (GT). The GT label is usually prepared for the retinal image by an experienced human observer to compare and identify the quantitative performance of the segmentation with the computer output. The comparison yields true positive (TP) (pixels detected as vessel pixels in Y and they appear as vessels in the GT label), false positive (FP) (pixels classified as vessel pixels in Y while they exist in the background in the GT label), true negative (TN) (pixels classified as background pixels in Y and they appear as non-vessel pixels in the GT label), false negative (FN) (pixels classified as background pixels in Y while they look as vessels in the GT label). Table 2 is showing the performance metrics used to compare the performance results.

Table 2 Performance metrics used in this work to compare the results.

Metric	Preprocessing threshold	
Accuracy	It measures the percentage of pixels correctly segmented in the dataset.	
	Accuracy=TP+TNTP+FP+FN+TN	
Sensitivity	SensitivityRecall=TPTP+FN	
Specificity	Specificity=TNTN+FP	
AUC	AUC is the area under the ROC curve it measures how perfect the method can distinguish whether the pixel is a vessel pixel or a background one (Vessel/background).	

Results

This section presents the quantitative performance results obtained by the proposed algorithm and the visual illustrations of the segmented retinal vasculature for best and worst case accuracy obtained in the retinal image datasets.

Quantitative performance results

For optimization, the empirical experiment is performed on each dataset training images like (DRIVE, STARE, and CHASE-DB1) using the parameter optimization (see Fig. 5). The Green channel is used in all experiments. Table 3 summarizes the achieved optimal results of specific parameters against each dataset as shown in the table (preprocessing threshold, filter threshold, and background artifact size). The selected values represent the best combination which indicates a very sensitive balance between the measurements that highlight a critical evaluation issue of any proposed solution. It is clear from the repeated empirical experiments applied to a different range of parameter values, that the performance of the given parameters are higher than the B-COSFIRE results with respect to sensitivity, specificity, and accuracy as summarized in Table 4.

Table 3 “Best combination” summary performance results.

	Parameters best combination	Optimized results	
Dataset	Prep. threshold	Filter threshold	Background artifact size	Sensitivity	Specificity	Accuracy	
DRIVE	0.3	30	48	0.790	0.971	0.955	
STARE	0.5	27	18	0.865	0.961	0.953	
CHASE-DB1	0.2	31	38	0.800	0.964	0.953	

Table 4 Summary results for the achieved improvement compared to the original B-COSFIRE results on DRIVE, STARE and CHASE-DB1.

Dataset	Approach	Sensitivity	Specificity	Accuracy	
DRIVE	Our approach	0.791	0.971	0.955	
B-COSFIRE	0.766	0.970	0.944	
STARE	Our approach	0.865	0.961	0.953	
B-COSFIRE	0.772	0.971	0.950	
CHASE-DB1	Our approach	0.800	0.964	0.953	
B-COSFIRE	0.759	0.959	0.939	

The bar charts in Fig. 6 are visualizing the comparison between B-COSFIRE method and optimized B-COSFIRE method with respect to sensitivity, specificity, and accuracy on DRIVE, STARE and CHASE-DB1 datasets.

Figure 6 Optimized B-COSFIRE vs. B-COSFIRE performance results on the datasets (A) DRIVE, (B) STARE and (C) CHASE-DB1.

Figure 7 DRIVE dataset segmentation result: (A) original image, (B) ground truth and (C) segmented vessels.

Figure 8 STARE dataset segmentation result: (A) original image, (B) ground truth and (C) segmented vessels.

Figure 9 CHASE-DB1 dataset segmentation result: (A) original image, (B) ground truth and (C) segmented vessels.

Segmented vasculature

Figures 7–9 are showing the segmentation results of DRIVE, STARE, and CHASE-DB1 datasets respectively. Two images from each dataset are selected with their best and worst case accuracies for visual results.

Quantitative results with optimal parameter selection

Our approach is to optimize and select the parameters that obtained the higher performance measures as compared to the B-COSFIRE method. The selected best parameters combination is pointed and underlined in Tables 5–7 for DRIVE, STARE, and CHASE-DB1 respectively. The results are sorted by sensitivity in descending order.

Discussion

This Work has improved the results of B-COSFIRE method compared to recent work in B-COSFIRE, as it has been improved in eight out of nine performance measures, as illustrated in Table 4, moreover ANOVA analysis has identified the parameter values that enhanced the segmentation output in DRIVE STARE and CHASE-DB1 in the three performance measures Acc. Sc. and Sp, and it showed that the parameters (preprocessing threshold, Filter threshold) have contributed significantly in this optimization, while the background artifact size contribution in the optimization is insignificant as detailed in the Tables 8–9, and finally the improvement is obvious in the Tables 10–12 when compared with other state of the art vessel segmentation results.

In this section, we have quantitatively identified the significant impact of each selected parameter on the vessel segmentation results using ANOVA inspections. The comparison of the optimized results with other methods is also discussed.

ANOVA inspections

For the improvement of the empirical experiment, one-way analysis of variance (ANOVA) is used to compare statistical significance of different values of each parameter on the results of the segmentation.

It is shown from Tables 8 and 9 that the threshold has a significant effect on the specificity (TN) for all the given datasets under study, whereas the background artifact size has no significant effect on the result for all of the given three datasets. Preprocessing Threshold, on the other hand, has a significant impact on all the results for the DRIVE and STARE dataset but not on the CHASE-DB1.

Table 5 Experiment-wise performance results on DRIVE dataset (the best parameters combination is underlined).

Experiment No.	Prep. threshold	Filter threshold	Background artifact size	Sensitivity	Specificity	Accuracy	
1	0.4	30	18	0.813	0.965	0.952	
2	0.4	30	28	0.811	0.966	0.952	
3	0.5	30	18	0.811	0.965	0.951	
4	0.4	31	48	0.798	0.969	0.954	
5	0.4	32	18	0.794	0.970	0.954	
6	0.3	30	28	0.794	0.970	0.954	
7	0.4	32	28	0.792	0.970	0.954	
8	0.5	32	18	0.792	0.970	0.954	
9	0.3	30	38	0.792	0.970	0.954	
10	0.3	30	48	0.790	0.971	0.955	
11	0.4	32	48	0.788	0.971	0.955	
12	0.5	32	38	0.788	0.971	0.954	
13	0.5	32	48	0.787	0.971	0.955	

Table 6 Experiment-wise performance results on STARE dataset (the best parameters combination is underlined).

Experiment No.	Prep. threshold	Filter threshold	Background artifact size	Sensitivity	Specificity	Accuracy	
1	0.5	31	18	0.802	0.964	0.948	
2	0.5	31	18	0.802	0.964	0.949	
3	0.5	31	28	0.801	0.963	0.951	
4	0.5	31	28	0.801	0.962	0.953	
5	0.5	27	18	0.800	0.961	0.953	
6	0.5	28	0	0.800	0.962	0.953	
7	0.5	28	9	0.799	0.963	0.953	
8	0.4	28	18	0.798	0.964	0.953	
9	0.3	30	18	0.786	0.964	0.952	
10	0.2	31	0	0.784	0.964	0.952	
11	0.2	31	9	0.784	0.964	0.952	
12	0.2	41	28	0.779	0.964	0.952	
13	0.1	41	38	0.779	0.964	0.952	

Table 7 Experiment-wise performance results on CHASE-DB1 Dataset (the best parameters combination is underlined).

Experiment no.	Prep. threshold	Filter threshold	Background artifact size	Sensitivity	Specificity	Accuracy	
1	0.1	31	18	0.802	0.964	0.953	
2	0.2	31	18	0.802	0.964	0.953	
3	0.1	31	28	0.801	0.964	0.953	
4	0.2	31	28	0.801	0.964	0.953	
5	0.1	31	38	0.80.1	0.964	0.953	
6	0.2	31	38	0.800	0.964	0.953	
7	0.3	31	18	0.800	0.964	0.953	
8	0.3	31	28	0.799	0.964	0.953	
9	0.3	31	38	0.798	0.965	0.953	
10	0.4	31	18	0.786	0.964	0.952	
11	0.4	31	28	0.785	0.964	0.952	
12	0.4	31	38	0.784	0.965	0.952	

Different possible combinations of one way ANOVA analyses are conducted to specify the significant impact of each parameter on the results and the conclusions found as follows:

Table 8 Analysis of variance (ANOVA) summary for parameters significance.

Measure	Accuracy	Sensitivity	Specificity	
Dataset Parameter	DRIVE	STARE	CHASE -DB1	DRIVE	STARE	CHASE -DB1	DRIVE	STARE	CHASE -DB1	
Prep. threshold	✓	✓	χ	✓	✓	χ	✓	✓	χ	
Filter	
threshold	χ	χ	✓	χ	χ	✓	✓	✓	✓	
Artifact size	χ	χ	χ	χ	χ	χ	χ	χ	χ	
Notes.

Note: ✓Significant Factor if P-Value ≤ 0.05, χ Insignificant Factor if P-Value > 0.05.

Table 9 Details of one-way analysis of variance (ANOVA).

Data Set	Parameters	Acc.	Se.	Sp.	Conclusion	
DRIVE	Preprocessing threshold	✓0.000.	✓0.000	✓0.000	As p-value = 0.000 ≤ 0.05, then preprocessing threshold parameters (0.1,0.2,0.3, 0.4,0.5,0.6) are significant with respect to accuracy as well as Sensitivity and specificity	
DRIVE	Filter threshold	χ 1.000	χ 0.252	✓0.000	As p-value = 0.000 ≤ 0.05 only for Specificity then threshold parameter is only significant for Specificity in DRIVE DB	
DRIVE	Atifact size	χ 0.995	χ 0.852	χ 0.886	Insignificant, as p-value > 0.05	
STARE	Preprocessing Threshold	✓0.000	✓0.000	✓0.000	As p-value = 0.000 ≤ 0.05, the prepr-ocessing threshold parameters (0.1,0.2,0.3, 0.4, 0.5,0.6) are sigy with respect to accuracy as well as Sensitivity and specificity	
STARE	Filter threshold	χ 0.975	χ 1.000	✓0.003	As p-value = 0.000 ≤ 0.05 only for Specificity then threshold parameter is only significant for Specificity in Stare DB	
STARE	Atifact size	χ 0.999	χ 1.000	χ 0.992	Insignificant, as p-value > 0.05	
CHASE-DB1	Preprocessing Threshold	χ 0.801	χ 0.998	χ 0998	As p-value = 0.000 ≤ 0.05, therefore parameters (0.1,0.2,0.3, 0.4) pre- processing threshold (0.5,0.6) are significant with respect to accuracy y as well as Sensitivity and specificity	
CHASE-DB1	Filter threshold	✓0.000	✓0.000	✓0.000	As p-value = 0.000 ≤ 0.05 for all the three performance measures then threshold parameter is significant for accurate, sensitivity and specificity	
CHASE-DB1	Atifact size	χ 0.947	χ 0.967	χ 0.972	Insignificant, as p-value > 0.05	
Notes.

Note 1: (✓) Significant Factor if P-Value ≤ 0.05, (χ) Insignificant Factor if P-Value > 0.05.

Note 2: Values in the columns: Acc., Se., and Sp. represent the significance of the p-value of ANOVA statistical analysis for the corresponding combination (Dataset, parameter, Performance Measure).

Table 10 Vessel segmentation performance on DRIVE dataset.

Method Type	Authors	Se.	Sp.	Acc.	AUC	Time	
Unsupervised	This Work	0.790	0.971	0.955	–	8 S	
Azzopardi & Petkov (2013b)	0.766	0.970	0.944	0.961	10 s	
Zhang et al. (2016)	0.747	0.976	0.947	0.952	–	
Yin et al. (2015)	0.725	0.979	0.940	–	–	
Roychowdhury, Koozekanani & Parhi (2015)	0.740	0.978	0.949	0.967	2.5 min	
Fraz et al. (2012a)	0.715	0.976	0.943	–	5 min	
Supervised	Orlando, Prokofyeva & Blaschko (2017)	0.790	0.969	–	–	–	
Dasgupta & Singh (2017)	0.7691	0.9801	0.9533	–	–	
Fu et al. (2016)	0.760	–	0.952	–	1.3 s	
Strisciuglio et al. (2016)	0.778	0.970	0.945	0.960	–	
Li et al. (2016)	0.757	0.982	0.953	0.974	1.2 min	
Fraz et al. (2012c)	0.741	0.981	0.948	0.975	2 min	
Marín et al. (2011)	0.707	0.980	0.945	0.959	1.5 min	

1. The preprocessing threshold is having a significant impact on the results for DRIVE and CHASE-DB1 as well as the existence of impact of the threshold on the specificity, this indicates that error in setting the threshold value decreases the true negative rate.

2. Based on ANOVA analysis, background artifact size factor on improving the method is found statistically insignificant on the accuracy and sensitivity.

3. In CHASE-DB1 dataset preprocessing threshold is of insignificant impact and this could be due to high background artifacts in this dataset images while the threshold factor is of high impact for the CHASE-DB1 dataset and deviating from the right balance in the threshold impact directly all the confusion matrix metrics.

4. In CHASE-DB1 method the factor threshold is significant which has an impact on the results. Therefore, the highest value of the threshold, i.e., 49 results in higher accuracy, precision, and specificity.

5. For DRIVE and STARE datasets preprocessing threshold is a significant factor for accuracy, precision, and specificity. However, filter threshold factor is only significant for specificity.

Comparison with other methods

The performance measure (Sensitivity, Specificity and Accuracy) of the proposed approach is compared with the previously published methodologies in Tables 10–12 for DRIVE, STARE and CHASE-DB1 datasets, respectively. It is clear from these results that the proposed approach is giving better results as compared to the previous methods. Tables 10–12 are showing that our results are better than the other state of the art supervised and unsupervised vessel segmentation methods in terms of accuracy, sensitivity, specificity), and comparing the average time of processing one image our optimization approach outperform the majority of approaches.

Table 11 Vessel segmentation performance on STARE dataset.

Method type	Authors	Se.	Sp.	Acc.	AUC	Time	
Unsupervised	This work	0.858	0.961	0.953	–	8 s	
Azzopardi & Petkov (2013b)	0.772	0.970	0.950	0.956	10 s	
Zhang et al. (2016)	0.768	0.976	0.955	0.961	–	
Yin et al. (2015)	0.854	0.942	0.933	–	–	
Roychowdhury, Koozekanani & Parhi (2015)	0.732	0.984	0.956	0.967	2.5 min	
Fraz et al. (2012a)	0.731	0.968	0.944	–	2 min	
Al-Diri, Hunter & Steel (2009)	0.752	0.968	–	–	11 min	
Supervised	Orlando, Prokofyeva & Blaschko (2017)	0.768	0.974	–	–	–	
Strisciuglio et al. (2016)	0.805	0.971	0.953	0.964	–	
Li et al. (2016)	0.773	0.984	0.963	0.988	1.2 min	
Fu et al. (2016)	0.741	–	0.959	–	–	
Fraz et al. (2012c)	0.755	0.976	0.953	0.977	2 min	
Marín et al. (2011)	0.694	0.982	0.953	0.977	1.5 min	

Table 12 Vessel segmentation performance on CHASE-DB1 dataset.

Method type	Authors	Se.	Sp.	Acc.	AUC	Time	
Unsupervised	This work	0.800	0.964	0.953	–	8 s	
Azzopardi & Petkov (2013b)	0.759	0.959	0.939	0.949	10 s	
Zhang et al. (2016)	0.756	0.966	0.946	0.956	–	
Roychowdhury, Koozekanani & Parhi (2015)	0.762	0.957	0.947	0.623	2.5 min	
Fraz et al. (2012a)	0.722	0.971	0.947	0.971	–	
Supervised	Orlando, Prokofyeva & Blaschko (2017)	0.728	0.971	–	–	–	
Li et al. (2016)	0.751	0.979	0.958	0.972	1.2 min	
Fu et al. (2016)	0.713	–	0.948	–	1.3 s	
Fraz et al. (2012c)	0.755	0.976	0.953	0.977	2 min	

It is pertinent to mention that the quantitative performance measures (Sensitivity, Specificity and Accuracy) achieved by the proposed method are better than other state of the art supervised, and unsupervised vessel segmentation methods. We have considered multi-objective optimization, where the three performance measures have been optimized simultaneously. The specificity reported by Dasgupta & Singh (2017) is higher than the proposed method, however their reported sensitivity and accuracy are less than our work. Furthermore, (Dasgupta & Singh, 2017) has used the DRIVE database only for performance evlauation. The performance on more challenging datasets (STARE and CHASE-DB1) is not reported. Fu et al. (2016) work has reported higher accuracy values, but their reported sensitivity is less and they had not reported the specificity.

The approximated time required to segment one fundus image is 8 s when performed on a CPU running at 2,700 GHz with 16 GB of RAM on Windows 10 Operating system. Currently this proposed work is applied by means of Matlab 2017b, The presentation can be computationally enhanced more. As shown in Tables 10–12, the processing time of this work is slightly near the newly presented approaches, the time includes the processing of bar and half bar filters. It is worth to note that we have used single processor for running the proposed work for optimization and it will be accelerated by running its computational model using the GPU programming.

Conclusion and Future Work

B-COSFIRE is a generic algorithm for vessel segmentation in retinal fundus images. Optimization of the parameters gets more efficient vessel segmentation with higher accuracy. It can also be tuned to detect and recognize patterns in videos. In this work, we have introduced a mechanism which is improving the results of B-COSFIRE. By optimizing the suppressing mechanism for the filter input and output thresholds. Such idea outperformed B-COSFIRE reported vessel segmentation results. The optimized three parameters are preprocessing threshold, as well as post-processing threshold, and background artifact size are chosen for optimization. The analysis is done with the help of ANOVA to show the impact of each parameter on results and also to evaluate the significance and insignificance of these parameters. ANOVA analysis for the experiments performed is showing a significant impact of the first two parameters like “preprocessing threshold” and “filter threshold”, while the third parameter “background artifact size” showing the insignificant impact on the results. The empirical experiments have evaluated and identified the new parameters configurations on three datasets. The selection of these optimized parameters makes this work get better results than the normal B-COSFIRE algorithm. Optimization of the three parameters discussed has outperformed the standard B-COSFIRE sensitivity, specificity, and accuracy on DRIVE and CHASE-DB1. At the same time in the STARE dataset, the selected combinations achieved higher accuracy and sensitivity while the specificity performed closer to B-COSFIRE reported results. It indicates the fact that the optimization of preprocessing threshold and filter threshold are not the whole optimization story. Although they are significant as per the ANOVA analysis, other parameters like σ, ϕ, and ρ need to be optimized for better results.

Additional Information and Declarations

Competing Interests

Author Contributions

Data Availability

The authors declare there are no competing interests.

Sufian A. Badawi conceived and designed the experiments, performed the experiments, analyzed the data, contributed reagents/materials/analysis tools, prepared figures and/or tables, authored or reviewed drafts of the paper, approved the final draft.

Muhammad Moazam Fraz conceived and designed the experiments, analyzed the data, contributed reagents/materials/analysis tools, authored or reviewed drafts of the paper, approved the final draft.

The following information was supplied regarding data availability:

http://vision.seecs.edu.pk/optimized_bcosfire/.

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
