# Peer review of "Optimizing the trainable B-COSFIRE filter for retinal blood vessel segmentation"

_PeerJ, doi:10.7717/peerj.5855_

## Round 0.1 · original submission · Major Revisions

Your manuscript has not been sent for review yet. In my role as editor, I read the abstract of a manuscript to decide how to move on. Your abstract is so minimal that I cannot move on. Please read for example:
http://www.icmje.org/recommendations/browse/manuscript-preparation/preparing-for-submission.html. To help you a little, adress the following. There are dozens of retinal blood vessel segmentation methods already published, why your paper? What is the state of the art and what is the problem? Are you presenting anything novel? Why should this novel hypothesis help reduce the problem? Provide numbers (images, outcome, p-values etc) of your experiment. End with something like your novel solution improves over currently existing solutions. If the abstract is OK, be aware that the reviewers will carefully assess if the manuscript aligns with the abstract.

---

## Round 0.2 · Major Revisions

The reviewers and I agree that the work describes interesting scientific results. Please address the reviewers comments. In addition, I found the following issues:
1. The optimization should be done on a separate set not used in the evaluation of the performance of your method. Can you clearly indicate how that is ensured?
2. The results section now contains a paragraph on statistical analysis, please move to the methods section.
3. The methods section contains a hypothesis in your new method description. Please define your hypothesis at the end of the introduction.

Reviewer 1 ·

Basic reporting

1) The language is clear

2) The Literature is the introduction is old. There are many new new references available in the field.
for example:
Line 57:
Recently, a trainable template matching method for vessel segmentation is introduced (Azzopardi and Petkov, 2013b; Azzopardi et al., 2015; Cinsdikici and Aydın, 2009; Fraz et al., 2012a; Hari et al., 2017; Sofka and Stewart, 2006).
This is not recently! 2006, 2009 and even 2012 is too old for this field.

Action to be taken: Add new references to the introduction and related work

3) Many typos here and there.
for example:
Line 88:
complex scenes.4) Detection----> complex scenes, and 4) Detection
Line 103:
shown in 4. ----> what 4? Figure 4 you mean?
Line 104:
difference of Gaussian (DoG) ----> defined before.
Line 111:
(as shown in (1)) --->? Figure 1?
Line 154:
* The optimization---> What *?
Line 233:
point 5 is at separate line where the other points are combined in the previous paragraph.

and many more.

Action to be taken: English editor need to check and validate the language, grammar, spaces and commas.

4) The figures' resolution is low.

Action to be taken: Enhance the resolution to be high

5) Table 3 label is wrong. The authors mean "summary" not "summery"

Experimental design

The research question is clear, and the experiments are well defined as well.
However, the statistics are not very clear. For example, Table 8 should show "that the noise size has no significant effect on the result for all of the given three datasets,"
but where is the statistical numbers to show that the new results has significant differences with the previous results?
Did the authors study the difference in time, space, CPU, or any other factors?

All the results are displayed with no evidence of significant differences

Validity of the findings

The findings are there linked to the original question. I would suggest to have a paragraph about how to choose the value of the suggested parameters?

Reviewer 2 ·

Basic reporting

1. In abstract, “The approach results are better than the other state of the are methods used for vessel segmentation.” is wrong and not precise, your results are not better than lots of supervised methods.
2. The last sentence of abstract is wrong
3. The concept of CORF is wrong, which has been defined in Azzopardi and Petkov, 2012
4. In section 4, the method organized not clear, for example
(1) how would you optimize the three parameters?
(2)”… “threshold(t)” showed in (5)…” may be ““threshold(t)” showed in (6)”
(3) what is the meaning of the sentence “Initially, the three approaches are conducted using the images from the CHASE-DB1 dataset”
(4) in the last paragraph in page 5, the defined symbol “Y” is not used in the paper.
5. in the last paragraph in page 6, for obtain the result Table 3, How do you set each parameter?
6. in table 4 and table 10, are you sure that the result of STARE database is the same as the result of STARE database?

Experimental design

The performance index values in the paper should be less than 1

Validity of the findings

the reference of the expressions in this paper is nonstandard. for example
(1)In the second paragraph from bottom in page 3, some references are “as shown in 3” “as shown in 4”.
(2)In the second paragraph from bottom in page 4, the same references are “as shown in (3)” “as shown in (3)”

---

## Round 0.3 · accepted · Accept

I agree with the reviewers and consider the current manuscript ready for publication.

# Reviewer 1 ·

Basic reporting

Again. I believe that the abstract does not reflect the true value of the research.
"The approach results are
better than the other state of the art methods used for vessel segmentation." not quit correct.

Experimental design

The manuscript is much better than before. It is in a good shape for publication.

Validity of the findings

no comments

Additional comments

no comments

Reviewer 2 ·

Basic reporting

The paper presents a method of optimizing trainable B_COSFIRE filter for the retinal blood vessel segmentation. The manuscript get better results than the normal B-COSFIRE algorithm results, I suggest to publish this paper

Experimental design

no comment

Validity of the findings

no comment